# Lactulose Modulates the Structure of Gut Microbiota and Alleviates Colitis-Associated Tumorigenesis

**DOI:** 10.3390/nu14030649

**Published:** 2022-02-03

**Authors:** Keizo Hiraishi, Feiyan Zhao, Lin-Hai Kurahara, Xiaodong Li, Tetsuo Yamashita, Takeshi Hashimoto, Yoko Matsuda, Zhihong Sun, Heping Zhang, Katsuya Hirano

**Affiliations:** 1Department of Cardiovascular Physiology, Faculty of Medicine, Kagawa University, Kagawa 761-0793, Japan; kzohiraishimobile@gmail.com (K.H.); lixiaodong2021.cn@gmail.com (X.L.); yamashita.tetsuo@kagawa-u.ac.jp (T.Y.); hashimoto.takeshi@kagawa-u.ac.jp (T.H.); hirano.katsuya@kagawa-u.ac.jp (K.H.); 2Key Laboratory of Dairy Biotechnology and Engineering, Ministry of Education, Inner Mongolia Agricultural University, Hohhot 010018, China; 15754881670@163.com (F.Z.); sunzhihong78@163.com (Z.S.); hepingdd@vip.sina.com (H.Z.); 3Oncology Pathology, Department of Pathology and Host-Defence, Faculty of Medicine, Kagawa University, Kagawa 761-0793, Japan; mazdayoko@gmail.com

**Keywords:** inflammatory bowel disease, colorectal cancer, lactulose, gut microbiota

## Abstract

Lactulose, a galactose-fructose disaccharide, is made from the milk sugar lactose by heating or isomerization processes. Lactulose is proposed to modulate gut microbiota and thus expected to be beneficial in treating inflammatory bowel disease. In the present study, we investigated the therapeutic effect of lactulose on gastrointestinal inflammation and inflammation-related tumorigenesis in a mouse model of colorectal cancer as well as its effect on gut microbiota composition. Azoxymethane (AOM)/dextran sulfate sodium (DSS) model was used in this study. Lactulose treatment was performed by feeding 2% lactulose for 14 weeks. Stool samples collected at 4 time points were used for metagenomic analysis of the microbiota. Pathological analysis was performed 21 weeks after AOM injection. AOM/DSS increased the macrophage counts, inflammatory cytokine expression, colorectal tumorigenesis, and imbalance in gut microbiota composition, as evidenced by increased pathogen abundance (e.g., *Escherichia* and *Clostridium*). Lactulose significantly inhibited the inflammatory events, and ameliorated inflammation and tumorigenesis. The composition of the intestinal microbiota was also restored upon lactulose treatment, and lactulose reduced pathogen abundance and increased the abundance of *Muribaculum* and *Lachnospirac**eae*. Meanwhile, the pathways related to Crohn’s disease were downregulated after lactulose treatment. Our findings suggest that lactulose restores the structure and composition of the intestinal microbiota, mitigates inflammation, and suppresses inflammatory tumorigenesis.

## 1. Introduction

The number of the patients with inflammatory bowel disease (IBD), which mainly refers to ulcerative colitis and Crohn’s disease, has been increasing in recent years [1]. IBD is characterized by inflammation in the gastrointestinal tract, weight loss, and rectal bleeding [2]. IBD is mainly caused by immune system dysfunction that arises as a result of excessive secretion of CD4^+^ helper T cells and interleukin-17 [3]. Macrophages also play a critical role in the pathogenesis of IBD. They are mainly divided into two types, classically activated macrophages and alternately activated macrophages. Classically activated macrophages (CD68+) are characterized by the secretion of pro-inflammatory cytokines IL-1β, IL-6, IL-12, and TNF-α. Carcinogenesis-related processes, such as apoptosis, proliferation, and angiogenesis are executed by inflammation mediators such as TNF-α [4,5]. IBD is frequently associated with inflammation-related colorectal cancer. IBD-related colorectal cancer is responsible for approximately 2% of the annual mortality from CRC overall, but 10–15% of the annual deaths in IBD patients [1]. IBD-related CRC patients are also affected at a younger age than sporadic CRC patients and have a 5-year survival rate of only 50% [1]. It is thus crucial to identify clinical methods other than surgery to alleviate IBD and CAC.

CAC progression is influenced by intestinal bacteria in the mucosal layer [6,7]. A human weighing 70 kg harbors nearly 0.2 kg bacteria, i.e., approximately 40 trillion bacterial cells [8]. Many bacteria, especially *Bacteroides*, *Escherichia coli* and *Fusobacterium nucleatum*, are involved in CAC progression [9,10]. *Firmicutes* are a well-known major strain of lactic acid-generating intestinal bacteria. Colon tissues obtained in biopsy from normal and CAC patients exhibit differences in the diversity of intestinal flora [11,12]. The proportion of these microbes has been used as an index for determining the risk of CAC development [13,14]. Microbial flora interventions (probiotics, prebiotics, antibiotics, fecal microbial flora transplantation, genetic manipulation, etc.) have been put to practical use in the treatment of IBD. They are also expected to have a preventive and ameliorating effect on CAC.

Lactulose, a disaccharide composed of galactose and fructose, is a products of the milk heating process. Lactulose is commonly used as a food additive to improve taste and promote intestinal transit in most countries. Lactulose is also currently used for treatment of constipation, hepatic encephalopathy, and chronic kidney disease [15,16,17]. It is not metabolized by the digestive enzymes in humans and therefore directly reaches the inflamed region in the gastrointestinal tract [18,19,20]. Therefore, the use of lactulose in the treatment of gastrointestinal diseases has been investigated [21], and it has been shown to exert protective effects against IBD [22]. Treatment with the galacto-oligosaccharide derivatives of lactulose was also reported to ameliorate cancer progression and restore the microbiota diversity in a mouse model of inflammatory tumorigenesis induced by administering azoxymethane (AOM) and dextran sulfate sodium (DSS) [23]. Metabolization of galacto-oligosaccharides by the intestinal microbes may be related to their therapeutic effects. In vitro experiments have demonstrated that lactulose was metabolized by 35 species of intestinal bacteria [24] and enhanced the production of short-chain fatty acids by activating *Bifidobacteria* and *Lactobacillus* [18]. Individuals treated with lactulose exhibited changes in the diversity of intestinal flora, i.e., increased abundance of *Bifidobacterium*—which produce lactic acid-derived probiotics and decreased abundance of pathogenic microbes, such as *Bacteroides* and *Clostridium* [25]. It is therefore hypothesized that lactulose restores the structure of intestinal microbiota, thereby mitigating the inflammatory events and suppressing inflammation-related tumorigenesis.

The present study addresses this hypothesis by using a mouse model of inflammation-related tumorigenesis generated using the co-administration of AOM and DSS. The effects of lactulose on tumor development, inflammatory events—including fibrosis—inflammatory signaling, and the structure and composition of intestinal microbiota were investigated to obtain mechanistic insights into the therapeutic effects of lactulose.

## 2. Materials and Methods

### 2.1. Chemicals

AOM (A5486) was purchased from Sigma-Aldrich Co. (St. Louis, MO, USA). DSS (MW 36,000–50,000) was purchased from MP Biomedical Inc. (Santa Ana, CA, USA). Lactulose was procured from Sanwa Kagaku Kenkyusho Co., Ltd. (Nagoya, Japan).

### 2.2. Animal Experiments

The experimental protocol for the generation of the inflammation-related tumorigenesis model (AOM/DSS model) and the workflow for lactulose treatment is shown in Figure 1. The present study utilized 6-week-old female C57BL/6NCrSlc mice (SLC, Inc., Hamamatsu, Japan) [26]. The mice were acclimatized for more than one week to a 12-h dark-and-light cycle at room temperature (25 °C) with *ad libitum* access to food and water. Mice were divided into three experimental groups (*n* = 8 in each group), i.e., Control group, AOM and DSS administration group (AOM/DSS), and AOM/DSS with lactulose treatment group (AOM/DSS + lactulose). The control group was intraperitoneally injected with saline and fed normal chow and drinking water during a 21-week experimental period. The AOM/DSS group was intraperitoneally injected with AOM (12 mg/kg body weight) on day 0. One week later, DSS treatment was initiated by providing the mice with drinking water containing 2% DSS for 1 week. The 1-week DSS treatment was repeated 3 times with a 1-week interval in between (Figure 1A). The body weight was measured weekly (Appendix A). In the AOM/DSS + lactulose group, mice were fed 2% lactulose-containing food for 14 weeks starting 1 week after the last DSS administration (Figure 1A). The food consumption was measured every 2 weeks during lactulose treatment, the dosage of lactulose was about 2.4~2.6g/kg (Appendix A). The dosage of lactulose was approximately 3 g/kg BW/day, based on the daily consumption of 3–4.5 g food and 20–30 g body weight.

Stool samples were collected at the time of AOM administration (T0), and 6 weeks (T1), 12 weeks (T2), and 21 weeks (T3) after AOM administration (Figure 1). The consistency of the stool samples was scored using the following classification: 0 = normal; 1 = slightly loose stool; 2 = severely loose stool; and 3 = diarrhea. The samples were then stored at −80° C until DNA extraction. After 21 weeks of AOM treatment, mice were euthanized by injecting pentobarbital (300 mg/kg, i.p.), followed by cervical dislocation, and the intestinal tract between the cecum and anus and spleen were excised *en bloc*. The length of the intestinal tract between the cecum and anus was measured. Then, the intestinal tract was longitudinally opened and the number of polyps was counted. Parts of the colon were placed in RNAlater^TM^ (Invitrogen, Carlsbad, CA, USA) for RNA extraction or lysed in liquid nitrogen for protein extraction, both being stored at −80 °C until the next process. The remaining intestinal tract was fixed on a rubber plate with a pin and immersed in formalin for 24 h. The spleen was weighed and fixed in 4% phosphate-buffered paraformaldehyde (Nacalai Tesque Inc., Kyoto, Japan) for 24 h and permeabilized with 70% ethanol. Cecal contents were obtained by squeezing with tweezers, and then weighed and stored at −80°C for short chain fatty acid analysis.

### 2.3. Institutional Review Board Statement

All animal experiments were approved by the Animal Care and Utilization Committee of Kagawa University, Japan (approval number: 19652). The animals were cared for in accordance with the institutional guidelines and the Guidelines for Proper Conduct of Animal Experiments.

### 2.4. Histopathological Evaluation

Tissues were fixed in 10% neutral buffered formalin overnight and then routinely embedded in paraffin. Tissue sections (3-micrometer thick) were subjected to hematoxylin and eosin (HE), Masson’s trichrome (MT), and immunohistochemical staining.

Histological scores of the HE-stained specimens were evaluated by a pathologist in a blinded manner at ×200 magnification. The score for inflammation (I) is as follows: 0 = none; 1 = mild; 2 = moderate; and 3 = severe. The score for extent of tumor (E) is as follows: 0 = none; 1 = mucosa; 2 = mucosa and submucosa; and 3 = transmural. The score for tissue regeneration (R) is as follows: 0 = complete regeneration; 1 = almost complete regeneration; 2 = regeneration with crypt depletion; 3 = surface epithelium not intact; and 4 = no tissue repair. The score for crypt damage (C) is as follows: 0 = none; 1 = 1/3rd of the base damaged; 2 = 2/3rd of the base damaged; 3 = only surface epithelium intact; and 4 = entire crypt and epithelium lost. The score for the percentage area involvement (P) is as follows: 1 = 1–25%; 2 = 26–50%; 3 = 51–75%; and 4 = 76–100%. The total histological score was calculated as I + E + R + C + P [27].

The fibrosis score was evaluated as follows: 0 = no fibrosis; 1 = mild fibrosis (focal mucosal/submucosal collagen deposition without architectural distortion); 2 = moderate fibrosis (significant mucosal/submucosal collagen deposition with modest distortion of mucosal/submucosal architecture but without obscuring the mucosal/submucosal border); 3 = severe fibrosis (extensive mucosal/submucosal collagen deposition with marked architectural distortion obscuring the mucosal/submucosal border).

Immunohistochemistry was performed using an R.T.U. VACTASTAIN KIT (Vector Laboratories, Burlingame, CA, USA) and a DAB staining kit (DAKO, Hovedstaden, Denmark). Antigen retrieval was performed using citrate acid (pH 6) for 20 min at 100 °C. Endogenous peroxidase was quenched using 3% hydrogen peroxide for 10 min at 20–25 °C. Tissue sections were blocked by incubating in 2.5% normal horse serum prepared in phosphate-buffered saline for 30 min at room temperature, and then incubated with anti-CD68 antibody (1:100 dilution in blocking buffer) overnight at 4 °C. The sections were then incubated with the biotinylated pan-specific secondary antibody (VECTASTAIN kit/DAB staining kit) for 30 min at room temperature, and then with a streptavidin-peroxidase complex for 30 min at room temperature. DAB was dropped onto the slides and DAB development was terminated by rinsing with water. Thereafter, the tissue sections were counter-stained with hematoxylin.

### 2.5. Real-Time PCR

Total RNA was extracted from the colon tissue using a Tissue Total RNA Mini Kit (Favorgen, Ping-Yung, Taiwan), and reverse transcribed into cDNA using the Prime Script RT Master Mix (Takara Bio Inc., Shiga, Japan). Real-time PCR was performed using the Viia7 Real-Time PCR system (Applied Biosystems, Waltham, MA, USA). The PCR protocol was as follows: initial denaturation at 95 °C for 20 s, followed by 70 cycles of denaturation at 95 °C for 5 s and annealing/extension at 60 °C for 30 s. TaqMan Fast Advanced Master Mix (Applied Biosystems) was used to monitor the PCR products, with the following TaqMan probes (Applied Biosystems): IL-6 (Mm00446190_m1), IFN-γ (Mm01168134_m1), TGF-β1 (Mm01178820_m1), and TNF-α (Mm00443258_m1). Mouse GAPDH Control mix (Applied Biosystems) was used as an endogenous control.

### 2.6. Quantification of Short-Chain Fatty Acid in Cecal Contents

Cecal contents obtained at T3 timepoint (Figure 1) were used to quantify acetic acid, propionic acid, *n*-butyric acid, isobutyric acid, *n*-valeric acid and isovaleric acid. The analysis was outsourced to Techno Suruga Laboratory Co., Ltd. (Shizuoka, Japan). The concentration of short-chain fatty acids in the sample was measured by gas chromatography after extraction. System GC-FID (7890B, Agilent Technologies, Santa Clara, CA, USA), Column: DB-WAXetr (30 m, 0.25 mm id, 0.25 μm film thickness), and guard column: DB-WAXetr (5 m, 0.25 mm id, 0.25 μm film thickness) were used.

### 2.7. Metagenomic Analysis

Five mice were randomly selected from each group, and their stools collected at T0, T1, T2, and T3 were subjected to metagenome analysis. DNA from stool was extracted using the QIAamp Fast DNA Stool Mini Kit (QIAGEN, Hulsterweg, Germany) according to the manufacturer’s instructions.

A total of 1.0 μg DNA was fragmented to approximately 350 bp by sonication. The fragments were end-polished, A-tailed, and ligated with the full-length adaptor for Illumina sequencing with further PCR amplification. PCR products were purified using AMPure XP beads, and subjected to quality assessment with the Agilent 2100 Bioanalyzer and Qubit 2.0 (Agilent, Santa Clara, CA, USA). The samples were sequenced using an Illumina NovaSeq 6000 instrument (Illumina, San Diego, CA, USA). After sequencing, raw data were processed to remove any mice-derived sequences using a KneadData software (v 0. 7. 5; The Huttenhower Lab, Boston, MA, USA). The remaining high-quality reads were used for subsequent analysis.

### 2.8. Nucleotides Sequence Accession Numbers

The metagenomic sequence data set of fecal microbiotas has been deposited in the National Center for Biotechnology Information Sequence Read Archive database (Accession Number: PRJNA741607).

### 2.9. Taxonomy and Functional Annotation

The HUMAnN2 software tool suite (https://huttenhower.sph.harvard.edu/humann2, accessed on 7 January 2022) was used to analyze high-quality metagenomic shotgun sequences. MetaphlAn3 (v 3. 0. 2) was used to calculate the relative abundance of the microbial communities. Pathways were annotated using HUMAnM2 (v 0. 11. 1).

### 2.10. Statistical Analyses

Differences in numerical variables among groups were evaluated using analysis of variance (ANOVA), and the Tukey–Kramer test was used for multiple comparisons for all pairs. A value of *p* < 0.05 was considered significant.

Nonmetric multidimensional scaling (NMDS) based on Bray-Curtis distances was employed to assess the structure of microbiota (gut microbiota β diversity) in different groups of mice. Analysis of similarity (ANOSIM; 999 permutations) was used to detect significant differences between groups. Differences in numerical variables among groups were assessed using analysis of variance and the Wilcoxon test (in R packages, http://www.r-project.org/, accessed on 7 January 2022) was used to calculate the differences within and between groups. Significance was set at *p* < 0.05. Linear discriminant analysis effect size (LEfSe) was performed to identify the biomarkers before and after lactulose treatment using the default parameters. Graphs were plotted using R software (v 4.3.0, http://www.r-project.org/, accessed on 7 January 2022), GraphPad Prism (v 6.01, https://www.graphpad.com/scientific-software/prism/, accessed on 7 January 2022), and STAMP (v 2.1.3, https://beikolab.cs.dal.ca/software/STAMP, accessed on 7 January 2022).

## 3. Results

### 3.1. Lactulose Inhibited the Development of AOM/DSS Model–Related Pathology

In the AOM/DSS group without lactulose treatment, mice exhibited loose stool and diarrhea as well as annal hernia (Figure 1B,C). The score of stool consistency (Figure 1B), the score of abnormal and gross bleeding (Figure 1C), tumor number (Figure 1D), and the weight of the spleen significantly increased at time point T3 (compared to the control group) (Figure 1E). Splenomegaly indicated augmented immune response. The 14-week treatment with 2% lactulose (approximately 3 g/kg BW/day) significantly inhibited the increase in tumor number and spleen weight (Figure 1B–E).

### 3.2. Lactulose Treatment Mitigated the Inflammatory Events in AOM/DSS Model

Histopathological evaluation revealed that AOM/DSS caused damage to the crypt and mucosal layer in the non-tumor area (Figure 2A). HE staining of the tumor sections revealed histopathological features corresponding to colorectal cancer (Figure 2A, tumor). AOM/DSS significantly increased the inflammation score in the non-tumor area (Figure 2B). Lactulose treatment ameliorated these damages and suppressed inflammation (Figure 2A,B). AOM/DSS induced fibrosis (Figure 2C) and significantly increased the fibrosis score (Figure 2D) in the non-tumor area. Lactulose treatment mitigated fibrosis; however, it had no significant (*p* = 0.22349) effect on the fibrosis score (Figure 2C,D) The mucosal and submucosal infiltration of CD68-positive cells increased in the non-tumor and tumor areas in the AOM/DSS model (Figure 2E,F). Lactulose significantly suppressed CD68-positive cell infiltration of the mucosal and submucosal areas (Figure 2E,F).

### 3.3. Lactulose Inhibited Inflammatory Cytokine Upregulation in the AOM/DSS Model

AOM/DSS significantly increased the transcript-level expression of TNF-α, IL-6, and TGF-β1, but not INF-γ (Figure 3). Lactulose treatment significantly suppressed the upregulation of TNF-α and IL-6, while it did not have a significant effect on the TGF-β1 or INF-γ levels.

### 3.4. AOM/DSS Did Not Have Any Effect on the Short-Chain Fatty Acid Levels in the Cecal Stool

Short chain fatty acids—products of the intestinal flora—have been reported to not only serve as an energy source for the host, but also exert biological effects, including tumor-suppressing effects (1, 2). The concentrations of acetic acid, propionic acid, *n*-butyric acid, isobutyric acid, *n*-valeric acid, and isovaleric acid in the cecal contents were evaluated as indices of the production of short chain fatty acids by the intestinal microbes (Figure 4). AOM/DSS as well as lactulose did not have a significant effect on the concentrations of these acids in the cecal stool.

### 3.5. Recovery of the Composition of the Fecal Microbiota by Lactulose Treatment in the AOM/DSS Model

Bray-Curtis distances among different samples were calculated and the differences in fecal bacterial composition among the three groups were determined based on the results of NMDS analysis and ANOSIM test (Figure 5). Initially, there were no significant differences in the microbiota composition among the groups at time point T0 (*p* = 0.138, Figure 5A,E). After AOM/DSS treatment (T1), samples in the control group clustered away from the other groups (*p* < 0.05) (Figure 5B). The distance between the control and AOM/DSS + lactulose group was greater than that between the control and AOM/DSS groups (Figure 5B,E). After a 5-week treatment with lactulose (T2), the control group and AOM/DSS + lactulose groups were closer than at T1 (Figure 5C), and the control group and AOM/DSS + lactulose group were significantly separated as compared to the control and AOM/DSS groups (Figure 5E). After 14 weeks of treatment with lactulose (T3), the samples in AOM/DSS group were significantly separated from the control (*p* < 0.05), while no significant difference was observed between the samples in the control and AOM/DSS + Lactulose groups (*p* = 0.205) (Figure 5D,E).

### 3.6. Lactulose Modifies the Gut Bacterial Composition after AOM/DSS Treatment

As AOM/DSS and lactulose treatment significantly affected the composition of the microbiota, we explored the changes in gut microbiota composition in response to treatment. The Wilcoxon test was used to calculate the differences in the microbial composition between the control and AOM/DSS-treated groups (AOM/DSS and AOM/DSS + Lactulose groups) at T1, wherein five genera (marked with * in Figure 6A) and six species (marked with * in Figure 6B) showed significant changes (*p* < 0.05). The relative abundance of *Escherichia* and *Clostridium*—opportunistic gut pathogens*—*was markedly increased in the AOM/DSS-treated group (compared with the control group), while the relative abundance of *Muribaculum* and *Lachnospirac**eae* reduced in the AOM/DSS group (*Lachnospirac**eae* have been recognized as short-chain fatty acid producers). The relative abundance of bacteria in the gut of AOM/DSS models recovered at T2, as the abundance of *Lachnospir**aceae* substantially increased, while that of *Escherichia* and *Clostridium* decreased (Figure 6A). The number of genera (Figure 6A) or species (Figure 6B) that showed recovery from T1 to T2 was greater with lactulose treatment (AOM/DSS + lactulose) than that seen with the normal diet (AOM/DSS), suggesting that lactulose is more beneficial for recovery from AOM/DSS-induced gut microbiota disorder. Further, the effect of long-term administration of lactulose on the gut microbiome was evaluated based on the difference in bacteria between T2 and T3 (Figure 6C). The abundance of four species in three genera changed significantly, i.e., the abundance of *Lactobacillus intestinalis*, *Lactobacillus murinus*, *Bacteroides caecimuris* increased significantly, while that of *Mucispirillum schaedleri* decreased significantly.

### 3.7. Lactulose Alters the Functional Properties of Gut Microbiota in the AOM/DSS Model

Functional pathways were analyzed to obtain insights into the functional changes in the gut microbiota at different time points. Analysis of the 40 pathways that significantly differed between the control and AOM/DSS-treated groups (AOM/DSS and AOM/DSS + lactulose groups) revealed that the starch degradation III pathway (Appendix A, No. 16) and heme biosynthesis pathway (Appendix A, No. 2) were significantly upregulated, whereas amino-acid metabolism pathways, such as the L-methionine biosynthesis III pathway (Appendix A, No. 27) and the L-threonine biosynthesis superpathway (Appendix A, No. 40), were significantly downregulated. Standard deviations are listed in Appendix A. Starch degradation and amino acid metabolism are associated with gastrointestinal disorders such as IBD [28], while heme biosynthesis has been shown to contribute to bacterial virulence, promote inflammation, and worsen the prognosis of patients with IBD [29]. With respect to the effect of lactulose (Figure 7), heme biosynthesis (Figure 7, Nos. 3 and 4) and L-histidine degradation (Figure 7, No. 13) were downregulated, while the N-acetylglucosamine and N-acetylmannosamine superpathways and N-acetylneuraminate (Figure 7, No. 5) and L-1,2-propanediol degradation pathways (Figure 7, No. 8) were significantly upregulated after lactulose intake (*p* < 0.05 in comparison with control and AOM/DSS groups, Figure 7, Appendix A). The genes encoding intermediaries of the 1,2-propanediol metabolism pathway have been reported to be upregulated in Crohn’s disease and shown to be associated with the growth of pathogens in the gut [30]. These results indicate that AOM/DSS significantly affected the composition and functional pathways of gut microbiota, while lactulose improved the relative abundance and functional pathways of the microbiome, phenomena that may be related to relief from AOM/DSS-induced symptoms.

## 4. Discussion

Based on the prebiotics and osmotic laxative properties, lactulose has been used as a laxative in the treatment of constipation for over 50 years [31]. This study demonstrated for the first time that lactulose restored the impaired composition and function of gastrointestinal bacteria, in a colitis-associated tumor mice model. Further, lactulose reduced inflammation, decreased tumor number, and ameliorated the macrophage counts and pro-inflammatory cytokine expression in an AOM/DSS mouse model.

The stool consistency was not changed by lactulose treatment compared to that seen in the A/D group (*p* = 0.1146). The doses of lactulose used in the current study (2.4~2.6 g/kg BW/day in mice) does not induce diarrhea or laxative effect, partly because this dose (equivalent to 0.2 g/kg BW/day in humans according to “the human equivalent dose”) is lower than that used for the treatment of constipation in adult humans (0.43–0.65 g/kg). In addition, the development of symptoms of constipation is well recognized in patients with ulcerative colitis (UC). In such a case, lactulose may have an additional beneficial effect for mitigating constipation.

The β diversity analysis indicated that short-term intake of lactulose may result in the regaining of the balance of gut microbiota composition after AOM/DSS treatment, while long-term intake could be more conducive to a healthy change in microbiota composition. Numerous studies have reported that lactulose increases the populations of beneficial gut bacteria (*Bifidobacterium* and *Lactobacillus*), elevates the levels of advantageous metabolites, reduces populations of harmful gut bacteria (*Clostridia*), and lowers fecal pH [18]. In the present study, lactulose significantly decreased the abundance of *Escherichia coli* and *Clostridium perfringens*, while it increased the abundance of *Muribaculaceae*. *Escherichia coli* is known to promote CAC in mouse models by producing colibactin, a genotoxic molecule [32]. *Clostridium perfringens* has also been reported as a pathogen that can accelerate the progression of CAC [33]. *Muribaculaceae* and *Clostridium perfringens* are the major mucin monosaccharide foragers and they occupy the same niche in the gut. As the degradation pathway of N-acetylglucosamine was upgraded after lactulose intake, we speculate the subsequent greater abundance of *Muribaculaceae* resulted in the consumption of more N-acetylglucosamine and impeding of the colonization of *Clostridium perfringens* in the gut [34]. Lactulose also reversed the suppression of *Lachnospiraceae* abundance induced in response to AOM/DSS treatment. *Lachnospiraceae* are known to be anaerobic, fermentative, and chemoorganotrophic, wherein some display strong hydrolyzing activities, and utilize starch, inulin, and arabinoxylan. Generally, *Lachnospiraceae* are associated with the promotion of resistance to intestinal pathogen colonization and the production of short chain fatty acids and bile acid metabolism [35,36]. Short chain fatty acids are also reported to exert a tumor-suppressing effect [37]. However, in the present study, cecal short chain fatty acids were not found to differ between control, AOM/DSS, and AOM/DSS + lactulose groups. Cecal stool short chain fatty acids do not appear to contribute to the tumor-suppressing effect of lactulose. Lactulose is also known to promote the growth of lactic acid bacteria and *bifidobacteria,* and more specifically, *Lactobacillus* in the colon [18]. Congruently, the present findings indicate that the abundance of *Lactobacillus intestinalis* and *Lactobacillus murinus* was increased by lactulose. In contrast, the abundance of *Mucispirillum schaedleri*, which exhibited protective effects against *Salmonella* colitis in a mouse model [38], was suppressed by lactulose in the present study.

Carbohydrate metabolism plays an important role in bacterial function. In IBD patients, starch degradation and amino acid metabolism pathway were significantly affected. Therefore, dietary fibers, such as resistant starches, are proposed as a promising treatment for IBD [28]. The present study revealed similar changes in the starch degradation and amino acid metabolism pathway in AOM/DSS. Heme biosynthesis has been reported to induce gut dysbiosis, worsen colitis, and potentiate the development of adenomas in mice [29]. Lactulose downregulated the heme biosynthesis pathway. This may contribute to the therapeutic effect of lactulose in AOM/DSS. The genes encoding intermediaries of 1,2-propanediol synthesis are upregulated in patients with Crohn’s disease [30]. Importantly, lactulose upregulated L-1,2-propanediol degradation pathways in our study. In addition, the fact that N-acetylneuraminate degradation was upgraded after lactulose intake may indicate that the altered expression of cell surface glycans plays a critical role in cancer (e.g., resistance to apoptosis and enhanced proliferation) [39]. These effects may be relevant in the context of the therapeutic effects in AOM/DSS. Lactulose has the potential to contribute to the prevention and treatment of IBD-related tumorigenesis. However, the mechanism underlying this lactulose-mediated inhibition is unclear. The identification of these mechanisms will be a subject in the next stage of our investigation.

AOM/DSS mice were in a proinflammatory state, which is characterized by intestinal shortening, splenomegaly, and impaired condition of stool and anus, in addition to increased scores of inflammations and fibrosis, macrophage infiltration, and increased expression of inflammatory cytokines, such as TNF-α, IL-6 and TGF-β1. It is noteworthy that the increased macrophage counts and enhanced proinflammatory state is observed even in the non-tumor area. These observations are consistent with those in the previous reports [4,5]. TNF-α has been reported to be upregulated in AOM/DSS-treated mice, while its receptor, TNF-Rp55, is expressed on leukocytes in the colon and mucosal layer. Binding of TNF-α to TNF-Rp55 results in the upregulation of NF-κB and COX-2 [40,41]. IL-6 has been reported to contribute to inflammation and fibrosis, and to regulate the immune system and activate cancer-related signal transduction pathways, such as NF-κB and STAT3 [42]. Lactulose mitigated all these proinflammatory states. This anti-inflammatory effect as well as the anti-tumorigenesis effect of lactulose have been reported previously [43]. The present study is the first to correlate the effect of lactulose on AOM/DSS pathology with its effect on intestinal microbiota and their metabolic pathways.

## 5. Conclusions

In conclusion, lactulose is suggested to be a potential therapeutic agent for IBD and IBD-related tumorigenesis. Its therapeutic effects on inflammation and tumorigenesis are suggested to be attributable, at least in part, to the recovery of the structure and composition of microbiota, which are associated with the recovery of functional and metabolic pathways associated with the microbiota in the healthier subjects.

## Figures and Tables

**Figure 1 nutrients-14-00649-f001:**
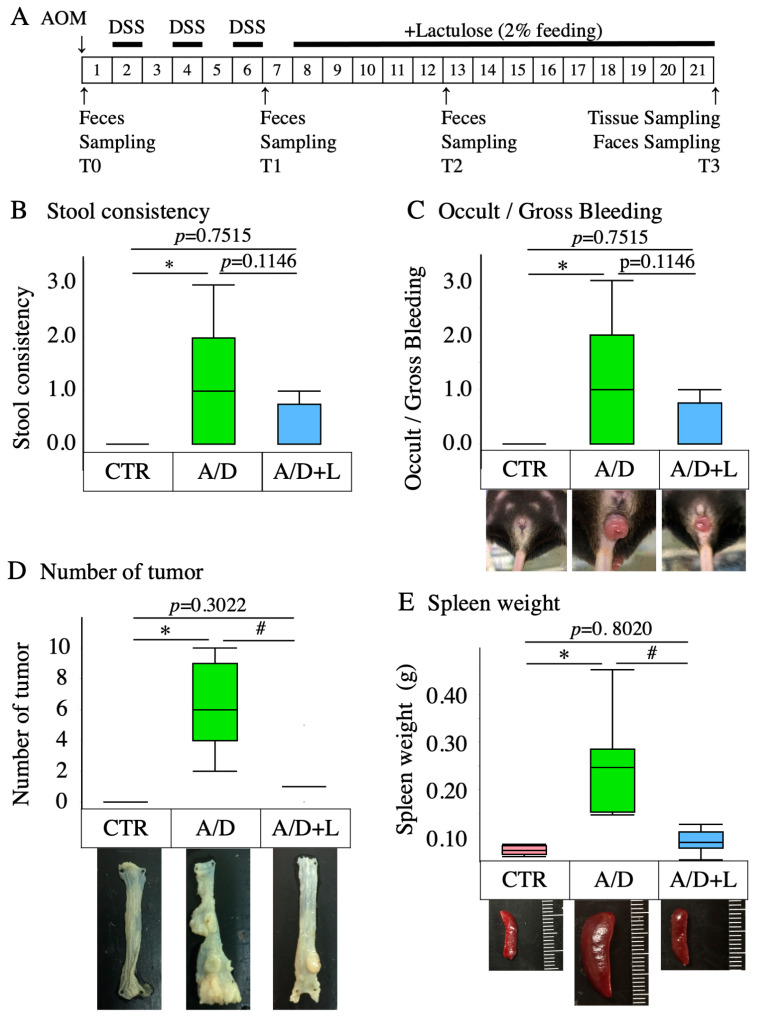
Inhibitory effect of lactulose on the development of AOS/DSS model-related pathology. (**A**) The experimental protocol for the generation of the AOM/DSS model (A/D) and subsequent treatment with lactulose (A/D+L) is shown. Mice were intraperitoneally injected with azoxymethane (AOM) (12 mg/kg body weight) on day 0. One week later, treatment with dextran sulfate sodium (DSS) was initiated by providing the mice 2% DSS in drinking water for 1 week. The 1-week DSS treatment was repeated 3 times with a 1-week interval. One week after the last treatment with DSS, lactulose treatment was initiated. The mice were fed lactulose-containing chow for 14 weeks. The dose of lactulose was approximately 2 g/kg weight/day. In the AOM/DSS group without lactulose treatment (A/D), the normal chow replaced the lactulose-containing chow. In the control group (CTR), administration of AOM, DSS and lactulose was omitted. Stool samples were collected on day 0 (T0) and 6 weeks (T1), 12 weeks (T2), and 21 weeks (T3) after AOM injection. All evaluations were performed 21 weeks after AOM injection. B–E: Summaries of the score of stool consistency (**B**), the scores of abnormal/gross bleedings (**C**), the weight of the spleen (**D**), and the number of tumors (**E**). Data are shown in boxplots (*n* = 8). * *p* < 0.05 vs. CTR, # *p*< 0.05 vs. AOM/DSS.

**Figure 2 nutrients-14-00649-f002:**
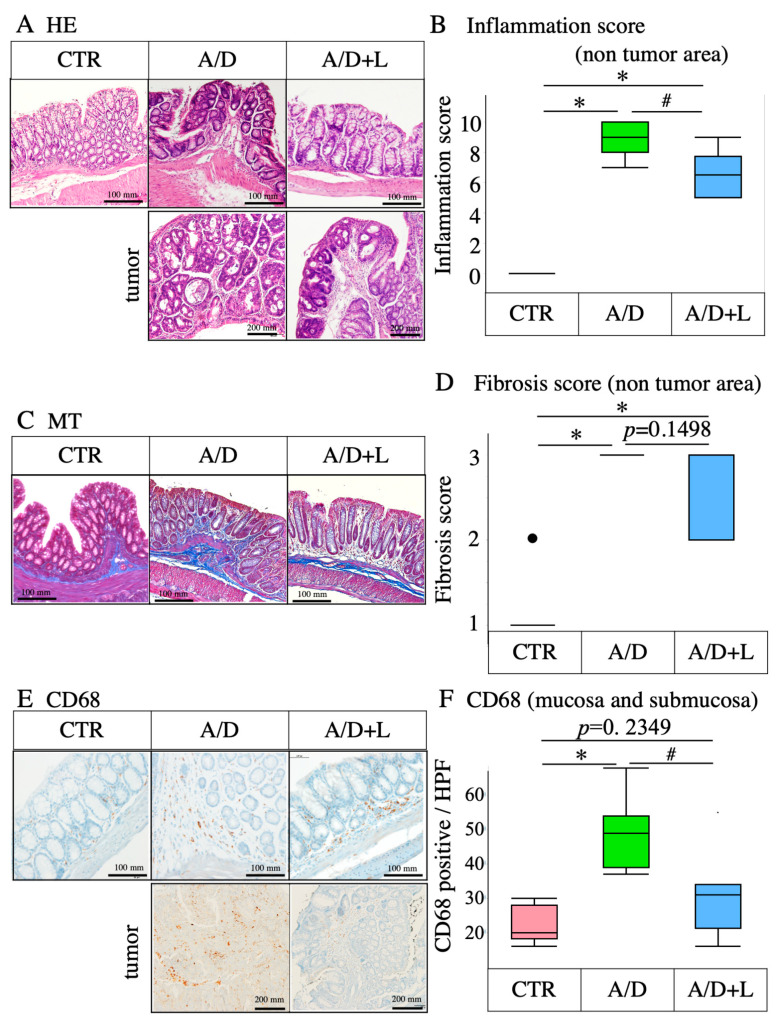
Inhibitory effects of lactulose on the inflammatory events in the AOM/DSS model. (**A**) Representative photomicrographs of HE staining of the tumor and non-tumor areas. (**B**) Inflammation score in the non-tumor area in the control (CTR), AOM/DSS (A/D) and AOM/DSS + lactulose (A/D+L) groups (*n* = 8). (**C**) Representative photomicrographs of MT staining. (**D**) Summary of the fibrosis score evaluated in the non-tumor area in the indicated experimental groups. (**E**) Representative photomicrographs of immunohistochemical detection of CD68 in the non-tumor and tumor areas. (**F**) The number of CD68-positive cells in the mucosal and submucosal layers of the non-tumor area per high power field (HPF: 0.196 mm^2^) in the indicated experimental groups. Data are shown in boxplots (*n* = 8). * *p* < 0.05 vs. CTR, # *p* < 0.05 vs. AOM/DSS.

**Figure 3 nutrients-14-00649-f003:**
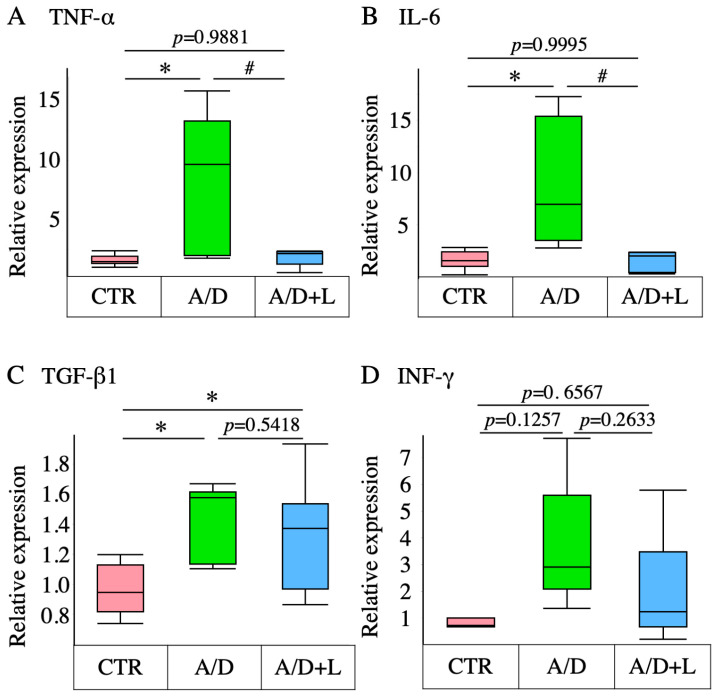
Effects of lactulose on the transcript-level expression of inflammatory cytokines in the AOM/DSS model. The transcript-level expression of tumor necrosis factor α (TNF-α) (**A**), interleukin 6 (IL-6) (**B**), transforming growth factor β1 (TGF-β1) (**C**), and interferon γ (INF-γ) (**D**) in the control (CTR), AOM/DSS (A/D), and AOM/DSS + lactulose (A/D+L) groups. Data are shown in boxplots (*n* = 8). * *p* < 0.05 vs. CTR, # *p* < 0.05 vs. AOM/DSS.

**Figure 4 nutrients-14-00649-f004:**
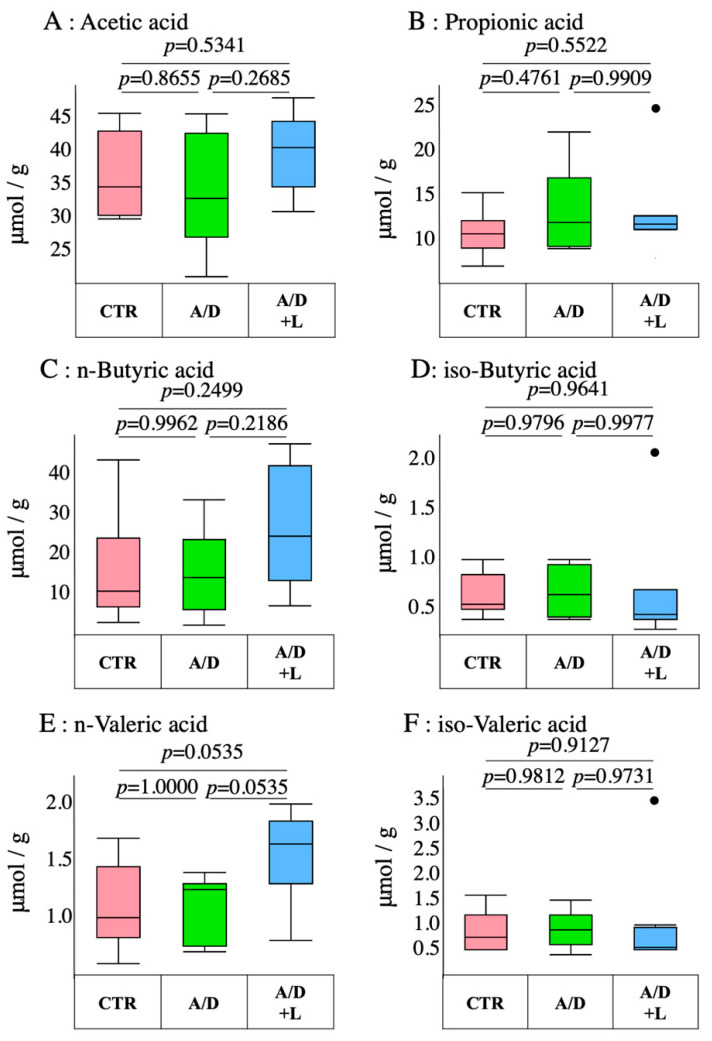
Analysis of short-chain fatty acid levels in the AOM/DSS model. The amounts of acetic acid (**A**), propionic acid (**B**), *n*-butyric acid (**C**), isobutyric acid (**D**), *n*-valeric acid (**E**), and isovaleric acid (**F**) in the cecal contents of the control (CTR), AOM/DSS (A/D), and AOM/DSS + lactulose (A/D+L) groups. Circles in the figure (**D**,**F**) are outliers, respectively. Data are shown in boxplots. (*n* = 8).

**Figure 5 nutrients-14-00649-f005:**
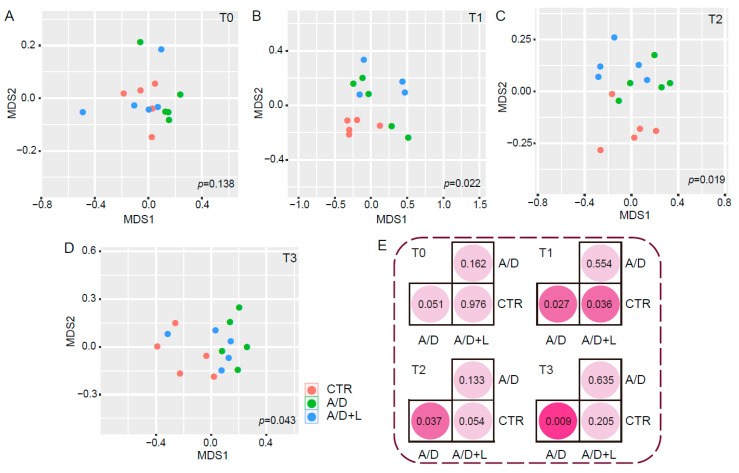
Effect of lactulose on the composition of the fecal microbiota in the AOM/DSS model. The scatter diagrams according to the feature values of MDS1 and MDS2 obtained after dimension reduction are shown for the control (CTR), AOM/DSS (A/D), and AOM/DSS + lactulose (A/D+L) groups (*n* = 5) at time points, T0 (**A**), T1 (**B**), T2 (**C**), and T3 (**D**). NMDS analysis based on Bray-Curtis distances revealed the feature values of each sample. The *p*-values were calculated among three groups (**A**–**D**) and between two groups via ANOSIM test (**E**).

**Figure 6 nutrients-14-00649-f006:**
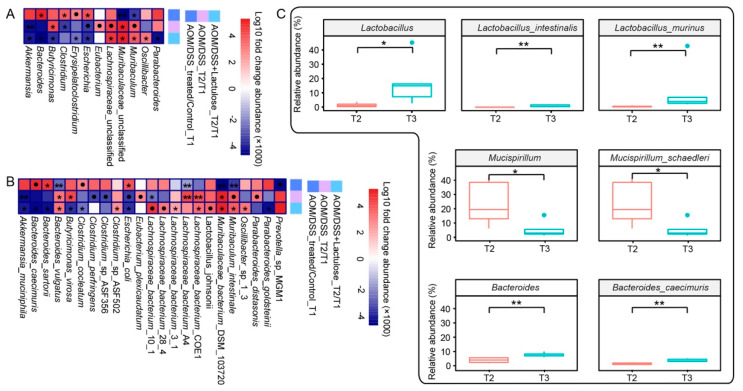
The effect of lactulose on the fecal microbial composition in the AOM/DSS model. (**A,B**) The heatmaps showing fold-change in the abundance of microbiota (genera in **A**, species in **B**) by the comparison indicated in a key: control vs. AOM/DSS at T1, T1 vs. T2 in AOM/DSS and AOM/DSS + lactulose. The differences in microbial composition were calculated by subtraction and significance was calculated using the Wilcoxon test. The values of log10 of the fold change (×1000) were scaled as shown in the key. ● 0.05 < *p* < 0.06, * *p* < 0.05, ** *p* < 0.01 (*n* = 5). (**C**) The relative abundance of the indicated genera and species at T2 and T3 in the AOM/DSS + lactulose group. * *p* <0.05, ** *p* < 0.01 (*n* = 5).

**Figure 7 nutrients-14-00649-f007:**
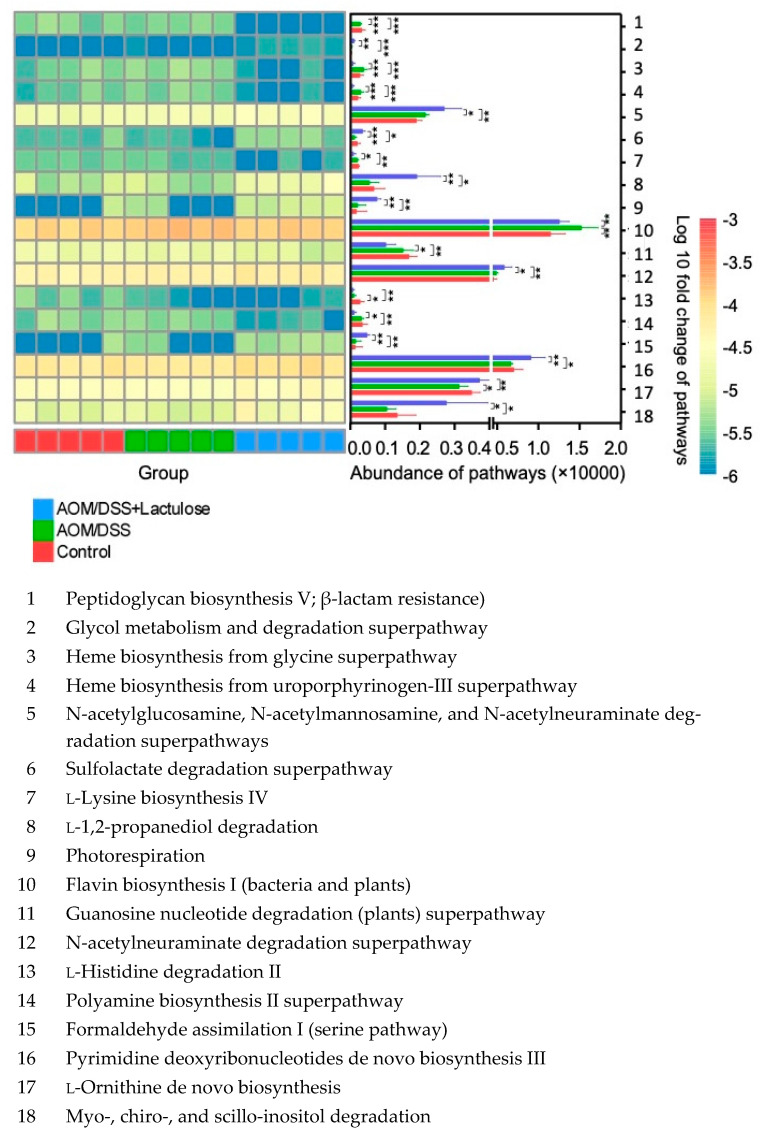
Differences in metabolic pathways among the three groups at T2. The top 18 pathways that showed significant (*p* < 0.01) difference between three group, i.e., Control, AOM/DSS, and AOM/DSS + Lactulose at T2. The abundance (copies per Million) of the pathways in each sample is shown in the heatmap and the average abundance of each pathway in each group is shown in bar chart. The significance of control vs. AOM/DSS, control vs AOM/DSS +lactulose, and AOM/DSS vs. AOM/DSS +lactulose groups analyzed using the *t*-test. * *p* < 0.05, ** *p* < 0.01, *** *p* < 0.001 (*n* = 5). Details are listed in Appendix A.

## Data Availability

The metagenomic sequence data set of fecal microbiotas has been deposited in the National Center for Biotechnology Information Sequence Read Archive database; the accession number is PRJNA741607.

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
