# Peer review of "Lactulose Modulates the Structure of Gut Microbiota and Alleviates Colitis-Associated Tumorigenesis"

_nutrients, 2022, doi:10.3390/nu14030649_

Round 1

Reviewer 1 Report

In the present study " Lactulose modulates the structure of gut microbiota and alleviates colitis-associated tumorigenesis"  Hiraishi et al. demonstrate a beneficial therapeutic effect of lactulose feeding on colitis-associated carcinogenesis in mice.
The design and performance of this study are of high quality and are interesting to the readers of Nutrients.

I have only some comments that have to be addressed before publication:

1) In real-time PCR experiments, the authors normalize their data to the housekeeper GAPDH. As it is known that GAPDH is highly regulated in tumors, the authors have to normalize their data to another housekeeper such as beta-actin or UBC.

2) The authors have to present data of food consumption and volumes of drinking water during the whole experiment to minimize misinterpretation of lactulose effects.

3) The authors should provide determined body weights during the experiment.

4) The authors should not compare A/D + L with the control group in statistical analyses, as the correct control group would be ctr + lactulose for this comparison.

Author Response

1) In real-time PCR experiments, the authors normalize their data to the housekeeper GAPDH. As it is known that GAPDH is highly regulated in tumors, the authors have to normalize their data to another housekeeper such as beta-actin or UBC.

Thank you very much for your comment. I agree with you that GAPDH may not be a good internal control in case of evaluation of tumor tissues. However, in the present study, we analyzed mRNA expression in the non-tumor areas. Therefore, we believe that GAPDH could serve as an internal control. In fact, the Ct values of GAPDH is consistent with different experimental condition. The row data is attached as below. Therefore, we made no change in the revised manuscript regarding this point.

2) The authors have to present data of food consumption and volumes of drinking water during the whole experiment to minimize misinterpretation of lactulose effects.

Thank you very much for your helpful comment. We added the following data of food consumption as the supplementary data. However, we did not keep the record of the consumption of drinking water.

Revised Figure S1B

Figure S1B: Food consumption of 3 groups from 8 weeks to 20 weeks.

Accordingly, we added the following statement in the Methods section: (L. XXX)

The food consumption was measured every 2 weeks during lactulose treatment, the dosage of lactulose was about 2.4~2.6g/kg (Figure S1B).

3) The authors should provide determined body weights during the experiment.

Thank you very much for your helpful comment. We added body weight data as below as the supplementary data.

Revised Figure S1A

Figure S1A: The body weight changes of 3 groups (n=8).

Accordingly, we added the following statement in the Methods section:

The body weight was measured weekly (Figure S1A). “

4) The authors should not compare A/D + L with the control group in statistical analyses, as the correct control group would be ctr + lactulose for this comparison.

Thank you very much for your helpful comment. I agree with you that the comparison of A/D + L group and the control is not appropriate to evaluate the effect of lactulose. However, the purpose of the comparison is just to know the degree of recovery from A/D model by lactulose treatment. Therefore, we compared the values of control and those of A/D model. As a result, we make no change in the revised manuscript regarding this point.

Reviewer 2 Report

This study aims to assess the benefit of lactulose as a treatment for IBD and to prevent colorectal cancer in IBD patients.  In the introduction it is stated that 2% of all colorectal cancer is associated with IBD but no reference is given and I do not think the incidence is as high as this.  Nonetheless it is a significant problem but the likelihood of lactulose being widely used in humans with IBD to treat IBD and prevent colorectal is low because of its laxative properties in humans.  Discussion of this needs to be included in the introduction and discussion.

The design of the study could be improved by including a lactulose only control group to document the effects on stool consistency and microbiota of lactulose alone in this mouse model.  It is very surprising that stool consistency was so much improved by lactulose (Fig 1B).  To interpret this and consider translation into humans it would be useful to know whether lactulose alone in these mice produces diarrhoea as this dose would be expected to do in humans.

Section 3.7 should be shortened and Table 1 and Fig 7 should be put in supplementary data.

In the Discussion it is stated that "Lactulose may be the key to inhibiting IBD-related tumorigenesis"  This is over-stating the significance of the findings and should be revised. 

The English needs further improvement.  It needs careful editing by someone expert in both science and English.  EG in Discussion line 395 I think inhibit is the opposite of what is meant and line 401 I think restored should be reversed.  The grammar is incorrect line 64 and line 66

Author Response

This study aims to assess the benefit of lactulose as a treatment for IBD and to prevent colorectal cancer in IBD patients.  In the introduction it is stated that 2% of all colorectal cancer is associated with IBD but no reference is given and I do not think the incidence is as high as this.  Nonetheless it is a significant problem but the likelihood of lactulose being widely used in humans with IBD to treat IBD and prevent colorectal is low because of its laxative properties in humans.  Discussion of this needs to be included in the introduction and discussion.

Thank you very much for your helpful comment. To avoid any misunderstanding, we revised the introduction by precisely citing the description of the reference [1]. Keller, D.S.; Windsor, A.; Cohen, R.; Chand, M. Colorectal cancer in inflammatory bowel disease: review of the evidence. Tech 507 Coloproctol 2019, 23, 3-13, doi:10.1007/s10151-019-1926-2.

Original introduction: L48-51

IBD-related colorectal cancer is responsible for approximately 2% of the annual colorectal cancer‒related mortality, and 10–15% of the annual deaths in case of IBD patients. IBD related colorectal cancer, i.e., colitis associated colorectal cancer (CAC) patients are also 50 affected at a younger age, and have a 5-year survival rate of 50% [1].

Revised introduction: L:47-51

IBD-related colorectal cancer is responsible for approximately 2% of the annual mortality from CRC overall, but 10-15% of the annual deaths in IBD patients [1]. IBD-related CRC patients are also affected at a younger age than sporadic CRC patients, and have a 5-year survival rate of 50% [1].

We added the following statements in the Discussion part as below. (L.402-409)

The stool consistency was not changed by lactulose treatment compared to that seen in the A/D group (p=0.1146). The doses of lactulose used in the current study (2.4~2.6 g/kg BW/day in mice) does not induce diarrhea or laxative effect, partly because this dose (equivalent to 0.2 g / kg BW/day in humans according to “the human equivalent dose”) is lower than that used for the treatment of constipation in adult humans (0.43-0.65 g/kg). In addition, the development of symptoms of constipation is well recognized in patients with ulcerative colitis (UC). In such a case, lactulose may have an additional beneficial effect for mitigating constipation.

The design of the study could be improved by including a lactulose only control group to document the effects on stool consistency and microbiota of lactulose alone in this mouse model.  It is very surprising that stool consistency was so much improved by lactulose (Fig 1B).  To interpret this and consider translation into humans it would be useful to know whether lactulose alone in these mice produces diarrhea as this dose would be expected to do in humans.

Thank you very much for your helpful comment. However, there seems to be some misunderstanding the effect of lactulose on the stool consistency shown in Figure 1B. There was no statistically significant improvement of the stool consistency (p=0.1146). In fact, the dosage of lactulose was approximately 2.4~2.6 g/kg BW/day in mice, equivalent to 0.2 g / kg BW/day in humans according to “the human equivalent dose”, which is lower than the clinical dose, 0.43-0.65 g/kg, used for the treatment of constipation treatment in adult humans.

We added this information for revised discussion part as below. (L.402-406)

The stool consistency was not changed by lactulose treatment compared to that seen in the A/D group (p=0.1146). The doses of lactulose used in the current study (2.4~2.6 g/kg BW/day in mice) does not induce diarrhea or laxative effect, partly because this dose (equivalent to 0.2 g / kg BW/day in humans according to “the human equivalent dose”) is lower than that used for the treatment of constipation in adult humans (0.43-0.65 g/kg).

Section 3.7 should be shortened and Table 1 and Fig 7 should be put in supplementary data.

In the Discussion it is stated that "Lactulose may be the key to inhibiting IBD-related tumorigenesis" This is over-stating the significance of the findings and should be revised. 

Thank you very much for your helpful comment. We moved table 1 to supplementary table. However, we believe that Fig.7 is an essential data. Therefore, we would like to leave this figure in the main text.

The description of the Section 3.7 was shortened by move table 1 to supplementary table.

Regarding the overstatement of the proposal, we made the following amendment:

Original discussion L458

Lactulose may be the key to inhibiting IBD-related tumorigenesis.

Revised discussion L455-456

Lactulose has the potential to contribute to the prevention and treatment of IBD-related tumorigenesis.

The English needs further improvement.  It needs careful editing by someone expert in both science and English.  EG in Discussion line 395 I think inhibit is the opposite of what is meant and line 401 I think restored should be reversed.  The grammar is incorrect line 64 and line 66

Thank you very much for your helpful comment. We asked the English editing serve to obtain linguistic assistance for the revised version of the manuscript. Attached is the certificate for the English proofreading. The English editing were performed twice by EDITAGE, the professional English proofreading service.

Thank you very much for the specific suggestions for the linguistic improvement. The followings are our response:

Original L395: Clostridium perfringens has also been reported as a pathogen that can inhibit the progression of CAC [33].

Revised L419: Clostridium perfringens has also been reported as a pathogen that can accelerate the progression of CAC [33].

Original L401: Lactulose also restored the suppression of Lachnospiraceae abundance induced in response to AOM/DSS treatment.

Revised L425: Lactulose also reversed the suppression of Lachnospiraceae abundance induced in response to AOM/DSS treatment.

Original L64: Lactulose, a disaccharide composed of galactose and fructose, is a products of heating process of milk.

Revised L64-65: Lactulose, a disaccharide composed of galactose and fructose, is the product of the milk heating process.

Original L66: Lactulose is also currently used as a therapeutic agent for constipation, hepatic encephalopathy, and chronic kidney disease.

Revised L66-67: Lactulose is also currently used for treatment of constipation, hepatic encephalopathy and chronic kidney disease.
